# HmPEAR: A Dataset for Human Pose Estimation and Action Recognition

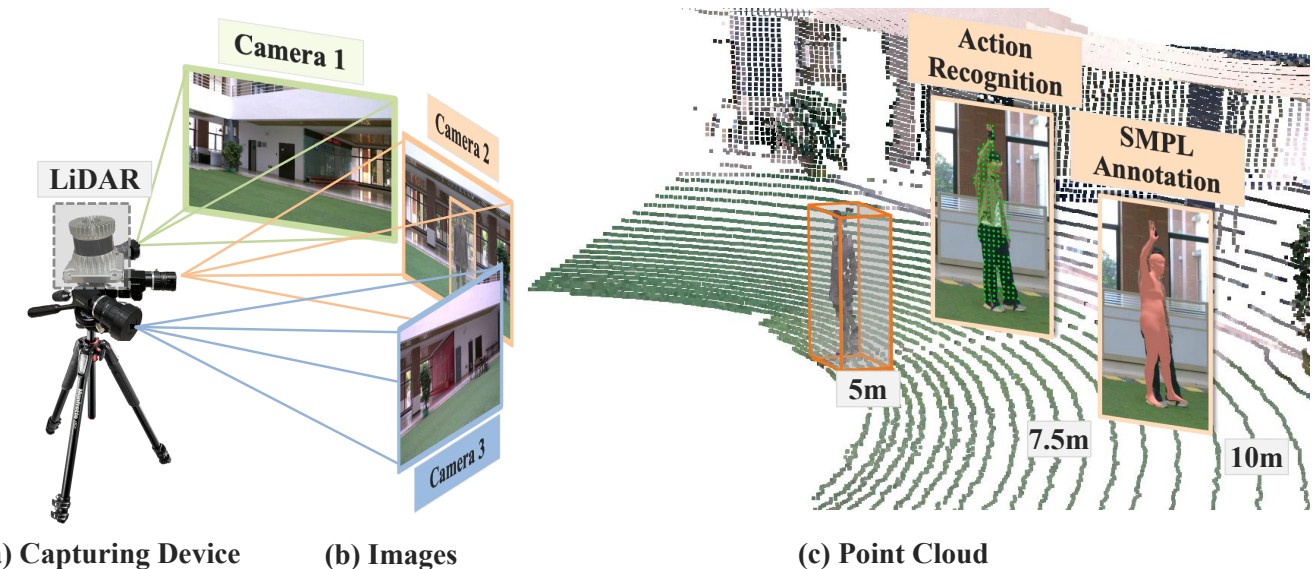

(a) Capturing Device  (b) Images  (c) Point Cloud

**Figure 1: (a) The capturing device, which is composed of a LiDAR and three RGB cameras. (b) Images from respective cameras. (c) Colored point cloud of a data collection scenario.**

## ABSTRACT

We introduce HmPEAR, a novel dataset crafted for advancing research in 3D Human Pose Estimation (3D HPE) and Human Action Recognition (HAR), with a primary focus on outdoor environments. This dataset offers a synchronized collection of imagery, LiDAR point clouds, 3D human poses, and action categories. In total, the dataset encompasses over 300,000 frames collected from 10 distinct scenes and 25 diverse subjects. Among these, 250,000 frames of data contain 3D human pose annotations captured using an advanced motion capture system and further optimized for accuracy. Furthermore, the dataset annotates 40 types of daily human actions, resulting in over 6,000 action clips. Through extensive experimentation, we have demonstrated the quality of HmPEAR and highlighted the challenges it presents to current methodologies. Additionally, we propose straightforward baselines leveraging sequential images and point clouds for 3D HPE and HAR, which underscore the mutual reinforcement between them, highlighting the potential for cross-task synergies.

**Unpublished working draft. Not for distribution.**

## CCS CONCEPTS

• **Computing methodologies → Motion capture**; **Activity recognition and understanding**; • **General and reference → Evaluation**.

## KEYWORDS

Dataset, Human Pose Estimation, Human Action Recognition

**ACM Reference Format:**

Anonymous Authors. 2024. HmPEAR: A Dataset for Human Pose Estimation and Action Recognition. In *Proceedings of the 32nd ACM International Conference on Multimedia (MM'24), October 28-November 1, 2024, Melbourne, Australia.* ACM, New York, NY, USA, 10 pages. https://doi.org/10.1145/nnnnnnn.nnnnnnn

## 1 INTRODUCTION

3D Human Pose Estimation (3D HPE) serves as the foundation for understanding human kinematics, offering detailed insights into the spatial configurations of body joints. This technology is fundamental to Augmented and Virtual Reality, simulations, and other related fields. On the other hand, Human Action Recognition (HAR) explores the dynamics of human movement, providing an advanced understanding of human behavior and intentions. The integration of these two domains, 3D HPE and HAR, enables a more comprehensive and nuanced interpretation of human actions. Therefore, creating a dataset that captures both 3D human poses and action categories is essential for in-depth research in human-centric studies.

Initial HPE efforts focused on 2D pose estimation [12, 16, 75]. However, the development of human body shape models [49, 65]

has allowed the representation of 3D human meshes with minimal parameters, facilitating the advancement of HPE from 2D to 3D [4, 37, 41, 46]. Nevertheless, reliance on RGB data presents limitations, such as sensitivity to lighting conditions and a lack of depth accuracy. While depth sensors like Kinect have offered insights into 3D HPE [5, 83], their application in outdoor or large-scale settings is limited. The rise of LiDAR, which, due to its precision and independence from lighting conditions, has been integrated into recent 3D HPE datasets [20, 27, 42, 77], facilitating advances in algorithmic development for 3D HPE [19, 42].

RGB-based HAR has seen extensive exploration, with approaches ranging from two-stream 2D CNNs [23, 31, 34, 61] to 3D CNNs [43, 68, 72] and transformers [2, 7, 48]. The current trends in RGB-based models are twofold: leveraging larger datasets or more complex models [2, 7, 44, 67], and pursuing multi-modal fusion to mitigate the shortcomings of single-modal approaches [24, 45, 56, 73]. Employing feature fusion methods to enhances the robustness of HAR, such as depth [22] and skeleton data [1, 11, 21, 25]. However, the majority of existing RGBD or RGB-Pose action recognition datasets neglect the intricate complexities inherent in dynamic, real-world scenarios. Previous efforts, such as those by Xu et al. [77], have made strides by incorporating LiDAR point clouds, yet they fall short in providing detailed 3D human pose annotations. This gap highlights an urgent need for a comprehensive dataset that integrates LiDAR point clouds and detailed human poses for HAR.

In this paper, we present a novel dataset, named HmPEAR, which integrates imagery and point cloud data for human 3D HPE and HAR. This dataset was captured using a high-precision 128-beam LiDAR and three high-resolution cameras (1920 × 1200 pixels). The data processing procedure involved manual tracking and further refinements. For the annotations of 3D human poses, we utilized professional motion capture (MoCap) equipment. Additionally, for action labels, we annotated them manually. To facilitate comprehensive evaluations for 3D HPE and HAR, we proposed three baseline models. Extensive experiments with proposed models and existing methods demonstrate the dataset's applicability and challenges. Our contributions are summarized as follows:

- We introduce a novel dataset that includes imagery and LiDAR point cloud with 3D human pose and action annotations. It bridges the gap between the fields of 3D HPE and HAR.
- Through extensive experiments, we have validated the quality and supplementary value of our dataset. These experiments also illustrated the challenges that our dataset poses.
- Additionally, we introduce a novel multi-task method that integrates 3D HPE and HAR, demonstrating their potential viability to complement each other.

## 2 RELATED WORKS

### 2.1 3D Human Pose Estimation Datasets

The Human3.6M [33] dataset represented a significant leap in RGB-based 3D Human Pose Estimation(3D HPE), offering extensive indoor scene data, while MPI-INF-3DHP [51] continued this trend, remaining limited to indoor settings. As for outdoor scenarios, 3DPW [70] introduced the first outdoor 3D HPE dataset with imagery, MuCo-3DHP [52] offered a collection of outdoor multi-person scenarios, featuring authentic interactions and occlusions.

However, they both lacked depth data, which is essential for full 3D context understanding. BEHAVE [9] introduced depth data using RGBD cameras, highlighting the critical role of depth information in such contexts. LiDARHuman26M [42] made strides by incorporating accurate LiDAR point cloud data. However, its potential was limited due to lower quality imagery and lack of synchronization. SLOPER4D [20] addressed these issues by enhancing annotation quality through refined optimization techniques. 3D HPE datasets often present challenges, as they predominantly feature human actions that are either performed casually or dramatically. This characteristic may lead models to learn poses in isolation without completely understanding the nuanced action semantics associated with those poses. As a result, there is a potential gap in comprehending the contextual meanings and intricacies of human movements within the datasets. This underscores the need for more comprehensive and context-aware 3D HPE datasets.

### 2.2 Human Action Recognition Datasets

Traditional Human Action Recognition(HAR) datasets have primarily consisted of RGB images, with a subset providing auxiliary depth maps and skeletal sequences. Early datasets such as KTH [58] and Weizmann [10] were limited in dataset size and scope. In recent years, various types of HAR datasets have emerged. UCF101 [62] and HMDB51 [39] have introduced RGB videos from diverse sources, covering a wide variety of action categories. Kinetics-400 [14] and Kinetics-600 [13] were developed to meet the demand for extensive data for effective model training. AVA [32] focuses on complex action detection and temporal localization, addressing real-world challenges in understanding human actions. On the other hand, NTU RGB+D [47, 59] pioneered to combination of RGB videos with depth maps and 3D skeletons, enriching datasets and enabling fusion-based HAR methods. BABEL [54] provides the dense action labels that are precisely aligned with their movement spans in the Motion Capture sequence. HuCentLife [77] encompasses diverse daily-life scenarios centering on people, assigning corresponding action labels to individuals in each scenario, and offering multi-modal data including imagery and point cloud. In the aforementioned multi-modal datasets, some lack point clouds, while others don't provide detailed 3D human poses. To address these gaps, there is a clear demand for a dataset that combines high-quality imagery and point cloud data with detailed 3D human poses and explicit action annotations.

### 2.3 3D Human Pose Estimation

Early 3D HPE efforts using RGB imagery for directly estimating joint locations [57, 66, 71] or body shapes [38, 69]. Advances in statistical human body shape models [49, 65] have facilitated the estimation of 3D human poses using a limited number of parameters [36, 37, 46, 53]. This enhances the robustness of joint location and shape predictions. Sophisticated optimization techniques have been introduced, such as HybriK [41], which decomposes body model parameter estimation, and NIKI [40], which leverages invertible networks to boost robustness and accuracy with RGB data. SPIN [37] uses the estimated 2D pose as initialization in SMPL iterative optimization to produce more accurate 3D poses and shapes. The incorporation of depth data [5] also helps the estimation. In

**Table 1: Comparisons with existing datasets. Depth refers to depth maps. Total frames denotes the count of valid frames referenced from previous works.**

| Dataset | PointCloud | RGB | Depth | SMPL | Bounding box | Subject | Scene | SMPL frames | Action class | Action clips | total frames |
|---|---|---|---|---|---|---|---|---|---|---|---|
| LiDARHuman26M [42] | ✓ | ✓ | ✗ | ✓ | ✗ | 13 | 2 | 184k | - | - | 184k |
| SLOPER4D [20] | ✓ | ✓ | ✗ | ✓ | ✓ | 12 | 10 | 100k | - | - | 100k |
| 3DPW [70] | ✗ | ✓ | ✗ | ✓ | ✓ | 7 | - | 51k | - | - | 51k |
| BEHAVE [9] | ✗ | ✓ | ✓ | ✓ | ✓ | 8 | 5 | 15k | - | - | 15k |
| CIMI4D [78] | ✓ | ✓ | ✗ | ✓ | ✓ | 12 | 13 | 180k | - | - | 180k |
| BABEL [54] | ✗ | ✗ | ✗ | ✓ | ✗ | 346 | - | 63k | 250 | 28k | 66k |
| SMART [15] | ✗ | ✓ | ✗ | ✓ | ✓ | 32 | - | 50k | 9 | 5k | 2,100k |
| HuCenLife [77] | ✓ | ✓ | ✗ | ✗ | ✓ | 65k | 32 | - | 12 | - | 6.1k |
| NTU120 [47] | ✗ | ✓ | ✓ | ✗ | ✗ | 106 | - | - | 120 | 114k | 8,000k |
| IKEA ASM [6] | ✗ | ✓ | ✓ | ✗ | ✓ | 48 | 5 | - | 23 | 16k | 3,046k |
| Ours | ✓ | ✓ | ✗ | ✓ | ✓ | 25 | 10 | 250k | 40 | 6k | 300k |

outdoor scenes, LiDAR [20, 42] caters to the need for long-range and more accurate depth. In our work, we combine RGB and LiDAR point clouds, leveraging the estimation of human 3D poses with HAR tasks to enhance the result of 3D HPE.

## 2.4 Human Action Recognition

Initially, the field of HAR has focused on integrating both temporal and spatial aspects. Baccouche.et al [3] combined the capabilities of 3D CNN with Long Short-Term Memory (LSTM) for spatiotemporal feature learning. SlowFast [30] is an early adopter of a dual-stream network architecture, capable of simultaneously capturing spatial and temporal features. Transformer-based methods [7, 44, 67] leverage attention mechanisms to capture long-range dependencies in temporal sequences. Recent developments in RGB-based HAR have introduced increasingly sophisticated strategies [17, 76], pushing the boundaries of learning-based HAR algorithms. State-of-the-art methods [1, 11, 25] tested on NTU RGB+D 120 [47, 59] suggest that combining RGB and pose information can yield promising results, which underscores the significance of pose data in such tasks. PoseConv3D [25] utilizes a 3D heatmap volume to represent human joints, while InfoGCN [17] employs attention-based graph convolution to capture the context-dependent intrinsic topology of human actions. Other approaches like STMT [82] and LocATe [63] directly leverage motion capture data to address the challenges in HAR. Recent LiDAR-based HAR methods have been developed to adapt to large-scale datasets and enhance accuracy. Among them, P4Transformer [28] employs the Transformer architecture to extract spatiotemporal features from continuous point clouds. PSTNet [29] was introduced to capture local structures and temporal dynamics. Mast-Pre [60] is presented as a self-supervised learning approach, leveraging spatiotemporal point tube masking and reconstruction tasks to glean structural information from point cloud videos. However, the absence of a comprehensive dataset restricts the development of unified models that are capable of capturing human motion and recognizing human actions in a more integrated manner.

## 3 DATA ACQUISITION

### 3.1 Device setup

As illustrated in Fig. 2, our device integrates a 128-beam Ouster OS-1 LiDAR and three Hikvision global shutter 1080p cameras. These

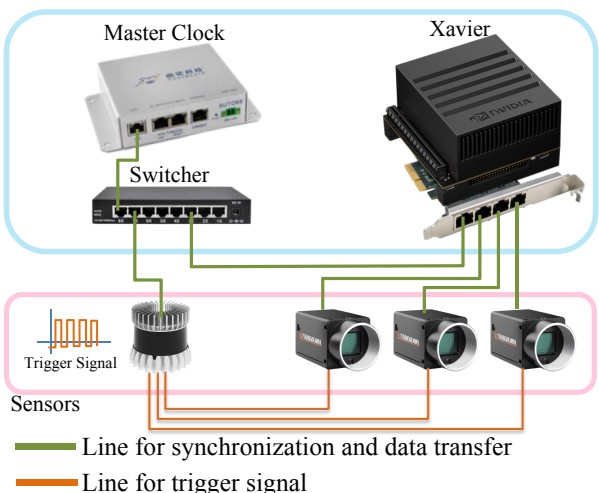

**Figure 2: The hardware setup of our capturing device. The LiDAR triggers the three cameras to capture images through trigger signals..**

sensors and an Auto66 master clock are connected to an NVIDIA Jetson AGX Xavier via a switch and a PCIe network card. The field of view (FOV) for each camera is 59.8°×46.2°, while the LiDAR's FOV spans 360°×45°. This setup provides an extensive FOV for cameras, facilitating data collection in scenarios of wide field. To eliminate distortion, global shutters are activated for all cameras. Fig. 1 demonstrates the data acquisition setup, with the capturing device mounted on a tripod. For 3D human pose acquisition, we utilize Noitom's Motion Capture (MoCap) system, PN Studio, which incorporates multiple Inertial Measurement Units (IMUs).

### 3.2 Calibration

We obtained the intrinsic parameters of the three cameras through Checkerboard Calibration [80, 81]. To determine the extrinsic parameters of different sensors, we followed the methodology outlined in Wen's work [74]. This involved using a terrestrial laser scanning (TLS) sensor to acquire a precise scene point cloud as a reference.

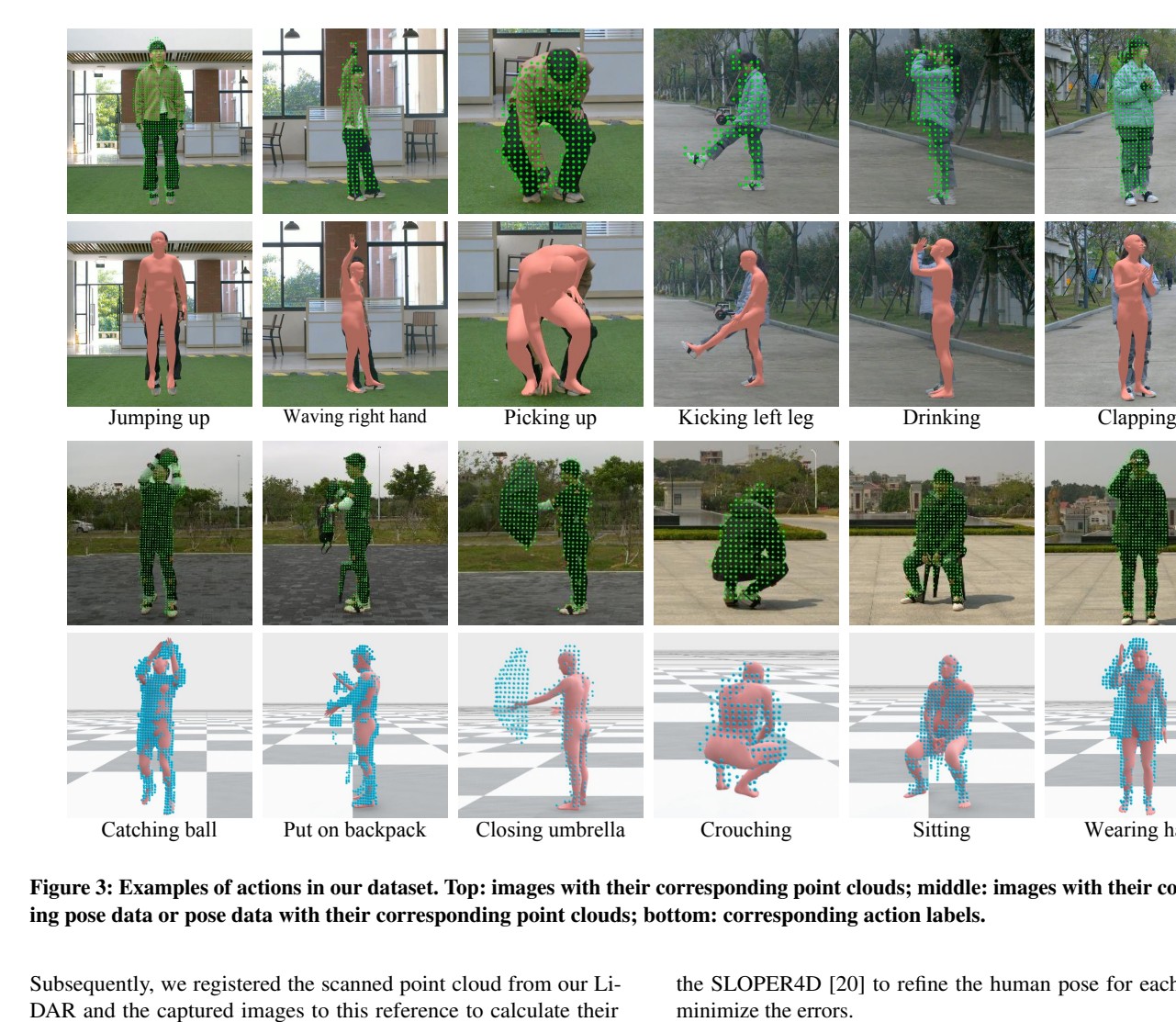

**Figure 3: Examples of actions in our dataset. Top: images with their corresponding point clouds; middle: images with their corresponding pose data or pose data with their corresponding point clouds; bottom: corresponding action labels.**

Subsequently, we registered the scanned point cloud from our Li-DAR and the captured images to this reference to calculate their relative transformations. Finally, we calculated the extrinsic parameters between all sensors using the relative transformations.

### 3.3 Time synchronization

To ensure sensor synchronization, it is crucial for all devices to operate under a unified timing mechanism and to capture data at specified times. We utilize the hardware-based Precision Time Protocol (PTP) to synchronize the sensors and the AGX Xavier with the master clock. Specifically, the Auto66 is configured as the grandmaster clock, the AGX Xavier serves as a boundary clock, and all sensors are configured as slave clocks. Furthermore, we engineered a simple circuit that triggers the three cameras to capture images at a pre-defined LiDAR rotation angle. Analysis of the PTP output verifies that the synchronization discrepancy among sensors is maintained below one millisecond. For the synchronization of 3D human pose data, we record jumping timestamps at the beginning and the end of each sequence and manually align it with the other data by detecting the peaks. Although his manual approach may slightly compromise synchronization precision, all sequences are optimized following

the SLOPER4D [20] to refine the human pose for each frame to minimize the errors.

### 3.4 Data collection

Fig. 1 illustrates a typical data collection scenario, where the subject performed actions with various positions and orientations. The distances from the capturing device to the subject typically range from 5 to 10 meters. To ensure realism, we did not impose restrictions on the details or duration of actions performed by the subjects.

### 3.5 Bounding box annotation

The ground of point clouds is eliminated by detecting the ground in the first frame and then applying K-Nearest Neighbors (K-NN) based abstraction to all frames. Subsequently, we employ the DB-SCAN [26] clustering algorithm to identify clusters within the remained point cloud. By selecting a single point within the cluster of the subject, we finally get the human point cloud. The 3D bounding box (bbox) is calculated based on the human point cloud and projected onto the images to get the corresponding 2D bbox. To ensure accuracy, all 2D bbox in the images are manually checked.

## 3.6 Action annotation

All action clips were annotated manually and subsequently reviewed for validation. For discrete actions, such as clapping, annotators were required to record the start and end of the action, ensuring completeness of the action clip. For continuous actions, like running, annotators were instructed to limit the duration of action clips to under five seconds, preventing the risk of excessive length.

## 3.7 3D human pose optimization

The SMPL [49] model is used to represent human poses. Each frame's pose $M_i$ is characterized by shape parameters $\beta$, joint orientations $\theta_i$ and transition $s_i$. The $\theta_i$ and $s_i$ are derived from human pose data recorded by Noitom's MoCap product, PN Studio. The $\beta$ are estimated by IPNet [8] using scanned models of subjects captured by iPhone's LiDAR. Let $Ver(M_i) \in \mathbb{R}^{6890 \times 3}$ represents the mesh vertices, $J(M_i) \in \mathbb{R}^{24 \times 3}$ represents the human joints. Followed SLOPER4D [20], several loss items based on Chamfer Distance (CD) and Mean Squared Error(MSE) are minimized to refine the 3D HPE annotations:

$$
\begin{aligned}
L_{all} =& \lambda_1 L_{CD}(Ver'(M_i), PC_i) \\
&+ \lambda_2 L_{MSE}((J(M_{i-1}) + J(M_{i+1}))/2, Jts(M_i)) \\
&+ \lambda_3 L_{MSE}((s_{i-1} + s_{i+1})/2, s_i) \\
&+ \lambda_4 L_{MSE}((\theta_{i-1} + \theta_{i+1})/2, \theta_i)
\end{aligned}
\tag{1}
$$

where $Ver'$ stands for visible vertices at the viewpoint of the LiDAR.

## 4 HMPEAR DATASET

HmPEAR is a novel dataset designed for 3D HPE and HAR. It encompasses over 300,000 frames of imagery, LiDAR point cloud, and bounding boxes of the subject, with more than 250,000 frames of corresponding 3D human poses. The dataset incorporates 40 daily actions performed by 25 subjects, resulting in over 6,000 action clips. These actions cover a wide range of human activities, including basic whole-body movements such as standing, walking, sitting, and running, specific actions involving the legs, such as kicking with the left leg and jumping, hand-related gestures like clapping and waving, as well as interactive behaviors such as making phone calls, wearing hats, and carrying bags. Each action clip varies in duration from 1.5 to 5 seconds. The dataset comprises 10 different scenes under various lighting conditions. A subset of typical actions from our dataset is illustrated in Fig. 3.

Tab. 1 provides a comparison of our dataset with other relevant datasets. Unlike existing 3D HPE datasets, which lack action annotations, HmPEAR encompasses annotations of 40 daily actions. In contrast to existing HAR datasets, which are limited to small-scale or indoor scenarios, HmPEAR expands the scope to larger and outdoor environments. By providing LiDAR point clouds and 3D human pose annotations, HmPEAR enhances the dataset's utility and applicability in advanced 3D HPE and HAR research.

## 5 METHODOLOGY

We propose simple baselines, as depicted in Fig. 4 and Fig. 5, composed of a Multi-modality Encoder, a Temporal Encoder, and a Multi-task Header. The subsequent sections will provide detailed descriptions of these modules and the architecture of the baselines.

## 5.1 Model Details

*5.1.1 Multi-modality Encoder.* As illustrated in Fig. 4, we employ the backbone of HRNet [64] to extract features of varying resolutions from each image. The features are subsequently fused by upsampling and addition. PointNet++ [55] is used to extract point-wise features from the human point clouds. Then, an affine transformation, derived to crop the original image to the input image, is used to project the point cloud features to a new feature map. The imagery and point cloud features are concatenated and further processed through multiple convolutional layers within the Fused-encoder to generate frame-wise multi-modal features.

*5.1.2 Temporal Encoder.* To leverage temporal information, the multimodal features are fed into a bidirectional GRU [18], enabling the extraction of both frame-wise and segment-wise features. The frame-wise features are denoted as $\{F_1^{GRU}, F_2^{GRU}, \ldots, F_m^{GRU}\}$, where $m$ represents the number of frames within the input segment. The segment-wise feature is denoted as $H^{GRU}$, as shown in Fig. 4.

*5.1.3 Multi-task header.* The frame-wise features are used to estimate the 3D positions of the 24 joints of the SMPL model, denoted as $J_i = Linear(F_i^{GRU})$. For action recognition, we replicate each $F_i^{GRU}$ by 24 times and concatenate it with $J_i$, denoted as $F_i^{Cat} = Cat(F_i^{GRU}, J_i)$. The resulting features are then input into a Spatial-Temporal Graph Convolutional Network (ST-GCN) [79] to produce action labels, denoted as $\hat{Act} = ST\text{-}GCN(\{F_i^{Cat}\})$. The segment-wise features are utilized to predict the human shape parameters $\hat{\beta} = Linear(H^{GRU})$, and, along with the predicted human joints $\{J_i\}$, are input into HybrIK [41] to estimate human poses. Similar to the SMPL model, HybrIK represents human motions using a limited set of parameters: $\hat{M}_i = HybrIK(\beta, J_i, \phi_i)$. The parameter $\phi_i$ denotes the twist along the direction between connected joints.

## 5.2 Proposed Models

We evaluate three distinct models, including ***PEAR-Proj***, as shown in Fig. 4, along with two additional models presented in Fig. 5. In ***PE-Proj***, we exclude the action classification head, which is denoted as the 3D HPE module. In ***AR-Proj***, we omit the 3D HPE component, with the $H^{GRU}$ being directly input into a Linear layer for generating classification results, denoted as the simple Classifier. These models aim to investigate potential cross-task synergies between 3D HPE and HAR, thereby demonstrating the effectiveness of our approach and dataset.

## 5.3 Training Paradigm

*5.3.1 Model inputs.* As the length of clips in our dataset varies, we have chosen to cut them into segments for training. Frame-wise annotations are cut in the same manner, while clip-wise annotations are applied to the corresponding segments. Consider an action clip of length $n$: $C_{Action} = \{c_1, c_2, ...c_n\}$, let $m$ represent the segment length. The time stride between frames in this clip is $\lfloor \frac{n}{m} \rfloor$, resulting in clipped segments: $\{S_1, S_2, ...\} = \{\{c_1, c_{\lfloor \frac{n}{m} \rfloor}, ...c_{m \times \lfloor \frac{n}{m} \rfloor}\}, \{c_2, c_{\lfloor \frac{n}{m} \rfloor + 1}, ...c_{m \times \lfloor \frac{n}{m} \rfloor + 1}\}, ...\}$. For 3D human pose clips, we randomly set the time stride between frames in the segment between 1 and 3.

To fully leverage the advantages of various 3D HPE datasets, our model employs a sampling strategy for different types of

**Figure 4: The proposed PEAR-Proj model processes sequential images and point clouds as input. It extracts features using a Multimodal Encoder and a Temporal Encoder. Subsequently, 3D human poses and action categories are generated by a Multi-task Header.**

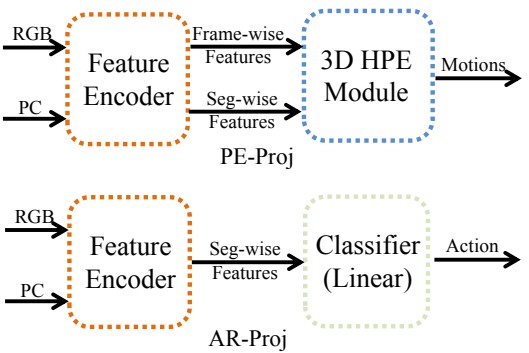

**Figure 5: PR-Proj comprises a Feature Encoder and a 3D HPE module. AR-Proj consists of the Feature Encoder and a simple classifier. The Feature Encoder itself is composed of the Multi-modality Encoder and the Temporal Encoder.**

datasets. Given the sampling rate $p_{act}$, a clip of $C_{action} = \{RGB, PointCloud, action\}$ is selected with probability $p_{act}$, while a clip of $C_{SMPL} = \{RGB, PointCloud, SMPL\}$ is chosen with probability $(1 - p_{act})$. For clips of $C_{SMPL}$, only the $\mathcal{L}_{pose}$ is calculated, and for $C_{action}$ clips, only the $\mathcal{L}_{act}$ is calculated.

*5.3.2 Loss function.* The overall loss function of our model is formulated as a weighted sum of 3D HPE loss $\mathcal{L}_{pose}$ and HAR loss $\mathcal{L}_{act}$, with the balance parameter $\eta$ as a hyperparameter.

$$\mathcal{L} = \eta \mathcal{L}_{pose} + (1 - \eta)\mathcal{L}_{act} \quad (2)$$

The action classification loss is computed using a cross-entropy loss method, which is simple yet effective.

$$\mathcal{L}_{act} = -\sum_{i=1}^{n} Act_i \log(\hat{Act_i}) \quad (3)$$

The 3D HPE loss is composed of the following items:

$$\mathcal{L}_{pose} = \lambda_{jts24}\mathcal{L}_{jts24} + \lambda_{jts17}\mathcal{L}_{jts17} + \lambda_{\beta}\mathcal{L}_{\beta} + \lambda_{\theta}\mathcal{L}_{\theta} \quad (4)$$

where $\mathcal{L}_{jts24}$ represents the loss associated with the 24 joints of the SMPL model, and $\mathcal{L}_{jts17}$ corresponds to the loss for the 17 joints defined by H36M, these joints are regressed using a fixed regressor $R_{h36m}$, given $\hat{M}_i = HybrIK(\beta, J_i^{24}, \phi_i)$, we obtain $\hat{J}_i^{17} = R_{h36m}Ver(\hat{M}_i)$. For $\mathcal{L}_{jts24}$ and $\mathcal{L}_{jts17}$, we employ the Mean

Squared Error(MSE) metric. Additionally, the human pose parameter losses, $\mathcal{L}_{\beta}$ and $\mathcal{L}_{\theta}$, are computed using the Mean Absolute Error(MAE) metric.

$$\mathcal{L}_{jts24} = \frac{1}{n}\sum_{i=1}^{n}(J_i^{24} - \hat{J}_i^{24})^2$$

$$\mathcal{L}_{jts17} = \frac{1}{n}\sum_{i=1}^{n}(J_i^{17} - \hat{J}_i^{17})^2$$

$$\mathcal{L}_{\beta} = |\beta - \hat{\beta}| \quad (5)$$

$$\mathcal{L}_{\theta} = \frac{1}{n}\sum_{i=1}^{n}|\theta_i - \hat{\theta}_i|$$

## 6 EXPERIMENTS

In this section, we detail the training settings and experimental results. We also provide detailed analyses of these experiments, aiming to assess the quality of our dataset and explore the cross-task synergies between 3D HPE and HAR.

### 6.1 Training Details

For comparison purposes, alongside HmPEAR, we employ 3DPW, SLOPER4D, COCO, H36M, and MPI-INF-3DHP [51] during the training of various methods. For 3DPW, COCO, H36M, AMASS [50], MPI-INF-3DHP, and SLOPER4D, we adhere to the official splits of the training and testing sets. Regarding HmPEAR, we employed different training and testing settings for 3D HPE and HAR. Firstly, we selected five subjects from different scenes as the testing set, denoted as $T_{test}$, and used the remaining data as the training set, denoted as $T_{train}$. Secondly, all data in $T_{test}$ and $T_{train}$ with action labels were used as the testing and training sets for HAR. Finally, 2 out of 5 subjects from $T_{test}$ and 8 out of 20 subjects from $T_{train}$ were used as the testing and training sets for 3D HPE. This setting for 3D HPE resulted in a subset of our proposed dataset of similar size as SLOPER4D.

For our proposed models, all images are cropped to a shape of 256x256 for input. The segment length, $m$, is set to 10. During training, we utilize a pre-trained HRNet. The initial learning rate is set to $2 \times 10^{-4}$ and is multiplied by a decay factor of 0.5 at epochs 2 and 5. We employ the Adam optimizer [35] for optimization. The training is conducted on two NVIDIA GeForce RTX 3090 GPUs for 15 epochs.

**Table 2: RGB-based 3D pose estimation. Models marked with * are cited from previous papers. All HybrIK models are trained including COCO, H36M, and MPI-INF-3DHP.**

| Method | Test / Train | 3DPW | | SLOPER4D | | Ours | |
|---|---|---|---|---|---|---|---|
| | | MPJPE↓ | PA-MPJPE↓ | MPJPE↓ | PA-MPJPE↓ | MPJPE↓ | PA-MPJPE↓ |
| VIBE | AMASS+3DPW | **83.2** | **52.0** | 100.0 | 61.5 | 85.5 | 47.0 |
| | AMASS+SLOPER4D | 124.3 | 66.8 | **86.6** | 52.4 | - | - |
| | AMASS+Ours | 122.9 | 60.9 | 107.8 | **49.6** | **63.2** | **39.8** |
| HybriK | +3DPW* | **71.3** | **41.8** | 75.8 | 50.0 | - | - |
| | +SLOPER4D* | 87.3 | 49.2 | **67.6** | **44.2** | - | - |
| | +Ours | 86.2 | 46.6 | 84.8 | 50.0 | **55.4** | **32.7** |

**Table 3: LiDAR-based 3D pose estimation. Performances marked with * are cited from previous papers.**

| Method | Test / Train | LiDARHuman26M | | SLOPER4D | | Ours | |
|---|---|---|---|---|---|---|---|
| | | MPJPE↓ | PA-MPJPE↓ | MPJPE↓ | PA-MPJPE↓ | MPJPE↓ | PA-MPJPE↓ |
| LiDARCap | LiDARHuman26M | 79.3* | 67.0* | 228.7* | 149.9* | - | - |
| | SLOPER4D | 212.3* | 128.3* | 86.1* | 65.1* | 211.4 | 110.8 |
| | Ours | 138.7 | 120.0 | 85.1 | 80.9 | 65.1 | 57.9 |

For the evaluation of 3D HPE, we employ the Mean Per Joint Position Error (MPJPE) and the Procrustes-Aligned Mean Per Joint Position Error (PA-MPJPE) as metrics. To assess action classification, we utilize per-clip and per-segment accuracy measurements.

## 6.2 Cross-Dataset Evaluation on 3D HPE

*6.2.1 RGB based 3D HPE.* We evaluate RGB-based 3D HPE methods, including VIBE and HybrIK, on 3DPW, SLOPER4D and our dataset(HmPEAR). The results are presented in Tab. 2. As VIBE requires AMASS for its motion discriminator, we trained all models from scratch with AMASS included. VIBE, trained on AMASS, HmPEAR, achieves better PA-MPJPE performance on SLOPER4D than the same model trained on AMASS, SLOPER4D. This underscores the quality and robustness of our proposed dataset. VIBE trained on {AMASS, HmPEAR} also yields better results when tested on 3DPW. However, this may be influenced by the domain gap between the datasets. Following the setting in HybrIK's original paper, we use all additional datasets including COCO, H36M, and MPI-INF-3DHP. Models trained on {+HmPEAR} demonstrate competitive performance.

*6.2.2 LiDAR Point cloud based 3D HPE.* We evaluate the LiDARCap [42] on LiDARHuman26M(LH26m), SLOPER4D, and our HmPEAR, the results are shown at Tab. 3. It is important to note that the LH26m employs a solid-state LiDAR for point cloud data collection, while both the SLOPER4D and our dataset utilize a mechanical LiDAR. This distinction may account for the performance decline when models trained on the LH26m dataset are tested on others, and vice versa. When comparing models trained on different datasets, LiDARCap exhibits limited generalization capabilities. Thought, the model trained on HmPEAR shows relatively better performance.

## 6.3 Evaluation of Proposed Models for 3D HPE

*6.3.1 Multi-modality based 3D HPE.* As shown in Tab. 4, when trained on the SLOPER4D dataset or our HmPEAR, PE-Proj demonstrated a degraded performance. This suggests that PE-Proj may be underperforming due to underfitting, as the SLOPER4D dataset or a subset of HmPEAR may not provide sufficient information for comprehensive learning. Notably, when SLOPER4D and our dataset were combined for training, PE-Proj exhibited improvement in performance and generalization across different testing sets. This underscores the supplementary value our dataset contributes to current 3D HPE datasets and the capabilities of the multi-modal model.

To investigate the contributions of different modalities, we further set the feature in PE-Proj of imagery or point cloud to 0, denoted as input type of 'PC' and 'RGB' in Tab. 4. Neither the PE-Proj(PC) nor the PE-Proj(RGB) could achieve comparable performance as the multi-modal model. The PE-Proj (PC) showed a more serious performance degradation, which we suppose should be attributed to the sparsity of the projected point cloud features.

*6.3.2 Multi-task based 3D HPE.* For PEAR-Proj, we recorded the optimal models for 3D HPE and HAR, labeled as 'BestPE' and 'BestAR' respectively, as presented in Tab. 4. The PEAR-Proj, when trained on HmPEAR, demonstrates results comparable to those of PE-Proj trained on both SLOPER4D and HmPEAR. This highlights the effectiveness of the cross-task synergies between 3D HPE and HAR. We further visualize several 3D HPE results in Fig. 6 (please zoom in for better visualization). The human SMPL models colored yellow represent the ground truth, while the human SMPL models colored blue are predictions by the PEAR-Proj.

## 6.4 Benchmark on HAR

We further evaluate several prevalent action recognition models on HmPEAR, including both RGB-based and point cloud (PC)

**Table 4: Comparisons of proposed models in 3D HPE.**

| Method | Inputs | Test \ Train | SLOPER4D MPJPE↓ | SLOPER4D PA-MPJPE↓ | Ours MPJPE↓ | Ours PA-MPJPE↓ |
|---|---|---|---|---|---|---|
| PE-Proj | RGB+PC | SLOPER4D | 112.6 | 80.7 | 194.7 | 95.9 |
| | RGB+PC | Ours | 178.1 | 110.6 | 129.5 | 97.2 |
| | RGB+PC | SLOPER4D+Ours | **55.5** | **38.9** | **59.9** | **43.0** |
| | RGB | SLOPER4D+Ours | 67.2 | 46.0 | 68.6 | 53.1 |
| | PC | SLOPER4D+Ours | 85.5 | 58.7 | 110.8 | 75.9 |
| PEAR-Proj(BestPE) | RGB+PC | Ours | 80.0 | 44.2 | 62.0 | 44.6 |
| PEAR-Proj(BestAR) | RGB+PC | Ours | 79.6 | 45.2 | 62.4 | 45.9 |

**Table 5: Benchmark on action recognition. PC: Point cloud data.**

| Method | Input | Test(Clip)↑ | Test(Seg)↑ |
|---|---|---|---|
| PSTNet | PC | 66.8 | 64.3 |
| P4-Transformer | PC | 65.8 | 63.9 |
| I3D | RGB | 57.6 | 55.5 |
| SlowFast | RGB | 63.7 | 62.2 |
| TimeSformer | RGB | 59.3 | 56.8 |
| Uniformer | RGB | 63.7 | 61.6 |
| AR-Proj | RGB+PC | 62.6 | 60.6 |
| PEAR-Proj(BestPE) | RGB+PC | 66.1 | 64.1 |
| PEAR-Proj(BestAR) | RGB+PC | **67.5** | **66.0** |

based methods. All RGB-based methods, including I3D [14], Slow-Fast [30], TimeSformer [7] and UniFormer [44] were pre-trained on the Kinetics-400 [14], utilizing a sequential image input of 8 frames of 224x224 pixels. The final classification layers of these methods are adapted for 40 categories. The results are presented in Tab. 5. PC-based methods show better performance than RGB-based methods, we infer that there is a domain gap between our HmPEAR and the Kinetics-400. Additionally, variations of lighting conditions in the HmPEAR posed greater challenges to RGB-based methods. The AR-Proj demonstrates comparable performances due to its multi-modal inputs but is still constrained by its simplistic classification head. With multi-modality inputs and multi-task output architecture, PEAR-Proj trained on HmPEAR achieved SOTA performance in Tab. 5.

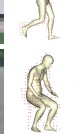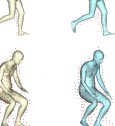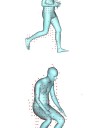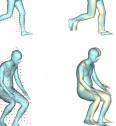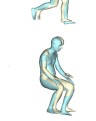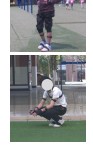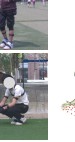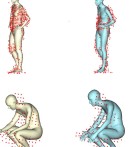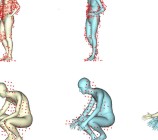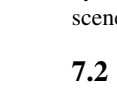

**Figure 6: Visualization of 3D HPE results of PEAR-Proj.**

### 6.5 Traning Analysis

We evaluate the performance of the PEAR-Proj at each epoch during training on our HmPEAR. The PA-MPJPE and the $Acc_{clip}$ are

recorded and depicted in Fig. 7. In the joint optimization of 3D HPE and HAR, PA-MPJPE shows a faster decrease in the initial stages. As the 3D HPE begins to yield plausible results, $Acc_{clip}$ increases more rapidly. 3D HPE and HAR are well entangled in PEAR-Proj.

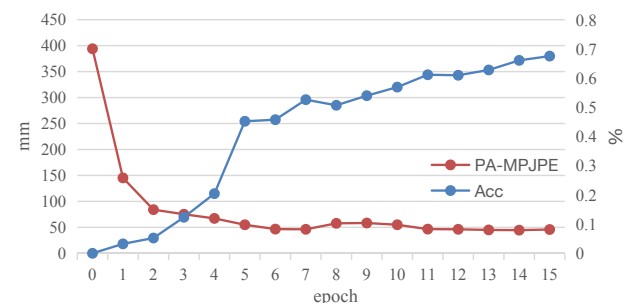

**Figure 7: The results of validation performed in each training epoch. Acc denotes the accuracy at the clip level.**

## 7 DISCUSSION

### 7.1 Limitations

It is important to acknowledge that the process of optimizing human poses and annotating actions in HmPEAR was resource-intensive. Consequently, we are unable to provide a large-scale dataset currently. Additionally, the age of subjects in our dataset falls within the range of 20 to 30, which may not provide sufficient diversity for training larger models. Furthermore, our dataset accounts for variations in time and lighting, but it does not yet include different weather conditions. In our future work, we plan to expand the dataset by incorporating a wider range of actions, more subjects, and diverse scenes with various weather conditions.

### 7.2 Conclusion

In this work, we introduce HmPEAR, a dataset designed to bridge the fields of 3D HPE and HAR. Through cross-dataset experiments, we have demonstrated the quality and robustness of our dataset. Furthermore, we proposed baseline models that incorporate multi-modal inputs and multi-task outputs. Comprehensive experiments and analyses showcased the ability of these models to leverage cross-task synergies between 3D HPE and HAR, offering a promising direction for future research on human pose and action analysis.

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
