# OpenReview forum: "HmPEAR: A Dataset for Human Pose Estimation and Action Recognition"
_acmmm.org/ACMMM/2024/Conference — MM2024 Poster_

### Official Review · Reviewer_2gB2 · 2024-05-07

**Rating:** 2
**Confidence:** 3

**Summary:**

The paper proposes a 3D human pose estimation and action recognition dataset . It further provides a baseline of 3D pose estimation and 3D human action recognition.

**Strengths:**

The paper provides an interesting dataset for a very challenging problem of 3D action recognition and pose estimation by providing a novel dataset and a baseline approach.

**Limitations:**

1) The paper is missing multiple ablation studies:
- For example, Section 5.1.1 mentions the fusion of 2D image features and 3D point cloud features using additional fusion encoder. What is the intuition mechanism behind concatenating the features using fusion encoder. What is performance if the features are fused simple using concatenation or addition?

2) What are the reason behind using the architectures PointNet++ and HRNet, when better models for example:
- point cloud feature extraction: PointNeXt (NeurIPS’22)
- and pose-estimation: TransPose (ICCV’21) and PCT (CVPR’23)
are already available.

3) It is still unclear the need for the proposed dataset as it only contains 25 subjects, whereas a large scale dataset like HuCenLife (ICCV’23) containing 65k subjects while including both RGB frames and 3D point clouds sample. Wouldn’t it be a good idea to take an existing dataset and generate the annotations?

4) The idea of doing a joint pose estimation and action recognition seems to be not well motivated in the paper.

5) Table 1 is very confusing as it is not easy to distinguished the output modalities and the metadata. It will be nice to take inspiration from Table 1 of paper: “Human-centric Scene Understanding for 3D Large-scale Scenarios” to distinguish among different components of the proposed and the existing datasets.

6) The method section of the paper is not clearly understandable:
- What is the intuition behind solving the 3D action recognition and 3D pose estimation jointly.
- It will be very helpful to mention the output dimension of HRNet and PointNet. If the output from the PointNet is reduced to simple channel, then it seems to be only retaining the global features. However, the same features are later used for “Seg-wise feature”. A little detailed information in this case will be very helpful in understanding the main idea behind this architecture.


7) The presentation of paper can be improved:
- Paper contains multiple abbreviation, which makes it difficult to understand. Consider replacing some abbreviations with their full forms.
- SMPL annotation are mentioned in the first page of the paper. However, the idea of SMPL was not clear before section 3.7. It would be nice idea to include a brief explanation in the introduction.
- AMASS dataset appear for the first time in the results section. If that is one of the similar data then why is it not included in Table 1 comparison.
- Is there any order or sorting dataset in Table 1? If yes, it is not clear enough.

8) In the result section,
- paper is missing a brief explanation of the evaluation metrics.
- paper is missing comparison with the SOTA. Please refer to the paper: “WHAM: Reconstructing World-grounded Humans with Accurate 3D Motion”. There are multiple methods performing better than the proposed approach on 3DPW dataset. Examples:
- TRACE: 5D Temporal Regression of Avatars with Dynamic Cameras in 3D Environments. In IEEE/CVF Conf. on Computer Vision and Pattern Recognition (CVPR), 2023
- Decoupling human and camera motion from videos in the wild. In IEEE Conference on Computer Vision and Pattern Recognition (CVPR), 2023.

**Suitability:**

2

---

### Official Review · Reviewer_4xci · 2024-05-25

**Rating:** 5
**Confidence:** 4

**Summary:**

This work proposes HmPEAR, a dataset for human pose estimation and action recognition, consisting of 300k frames collected from 10 scenes and 25 subjects. For each RGB video, there are corresponding LiDAR point clouds, 3D human poses, and action categories as annotations. The data is captured using one LiDAR and three RGB cameras with time synchronization. Bounding box annotations are obtained by performing DB-SCAN on the point cloud, action labels are manually annotated, and SMPL annotations are calculated using SLOPER4D. Cross-dataset experiments demonstrate that HmPEAR has better generalization capabilities for both RGB-based and LiDAR-based 3D pose estimation.

**Strengths:**

1. The tools for data collection and the data annotation pipeline are professional.
2. The dataset contains the highest number of annotated SMPL frames compared to previous datasets.
3. The overall experiments demonstrate that models trained on HmPEAR exhibit good generalization.
4. The paper is well-written and easy to follow.

**Limitations:**

## Dataset:
1. The number of total frames, scenes, action classes, and subjects is still limited compared to previous datasets. It would be beneficial to have more scenes and action classes in future versions of the dataset.
2. For the action class labels, it would be advantageous to include captions, such as "a person standing up then turns around." The authors could provide an official results of label-to-sentence augmentation (with LLM or human labor), for example, "carrying sth" can be augmented to many sentences, such as "a person is carrying an object." or "someone is holding and moving something." Such annotations can be helpful for CLIP-like models and text-to-motion tasks.
3. Will the data, data collection, and annotation pipeline be open-sourced?

## Evaluation:
1. In Table 2: RGB-based 3D pose estimation, the HybrIK evaluation results on the proposed dataset are missing(SLOPER4D and 3DPW), making it difficult to compare HmPEAR with SLOPER4D and 3DPW. It would be better to provide the results on HmPEAR instead of just citing from the previous paper.

**Suitability:**

3

---

### Official Review · Reviewer_iatG · 2024-05-25

**Rating:** 5
**Confidence:** 3

**Summary:**

This paper introduces a new dataset called HmPEAR to connect the field if 3D Human Pose Estimation (3D HPE) and Human Action Recognition (HAR). This dataset comprises imagery, LiDAR point cloud along with human pose and action annotations.  In addition, this paper introduces a mukti task combining the human pose estimation and human action recognition.

**Strengths:**

1. The paper is well structured and well written.
2. Introduction of a novel dataset always benefits the community.
3. The cross data evaluation on pose estimation proposes competitive results.

**Limitations:**

1. Baseline experiments seems limited.
2. Time/Frequency synchronisation needs to more clearly mention.
3. For the temporal encoders why not consider TCNs?
4. Did the authors consider using transformer encoders for the PEAR-Proj model?

**Suitability:**

3

---

### Official Review · Reviewer_RrkK · 2024-05-26

**Rating:** 3
**Confidence:** 3

**Summary:**

this paper introduces HmPEAR, a novel dataset crafted for advancing research in 3D Human Pose Estimation (3D HPE) and Human Action
Recognition (HAR), with a primary focus on outdoor environments.

**Strengths:**

1. dataset contribution is good, compared to other point clouds based benchmark, the proposed dataset has larger scale
2. the paper is well-written and clear

**Limitations:**

1. the proposed method is a bit complex, usually for a dataset paper, we want a simple yet efficient baseline, could author show a very naive version of method? (like point cloud encoder and keypoint regression decoder)
2. the results in Table 2 seems mean that the proposed dataset for training is not so useful, it makes contribution weak, could author explain more about the results?

**Suitability:**

2

---

### Meta-Review · Area_Chair_7zYe · 2024-07-03

**Recommendation:** Accept (Poster)
**Confidence:** 5

**Metareview:**

This paper proposes a novel multimodal dataset crafted for advancing research in 3D Human Pose Estimation and Human Action Recognition, with a primary focus on outdoor environments.
The paper is clearly written, the contributed dataset is large-scale and it has been carefully designed, the cross data evaluation on pose estimation proposes competitive results.
For all these reasons all reviewers agreed on accepting the paper and I also recommend acceptance.